# Seed Priming with Sorghum Water Extract Improves the Performance of Camelina (*Camelina sativa* (L.) Crantz.) under Salt Stress

**DOI:** 10.3390/plants10040749

**Published:** 2021-04-12

**Authors:** Ping Huang, Lili He, Adeel Abbas, Sadam Hussain, Saddam Hussain, Daolin Du, Muhammad Bilal Hafeez, Sidra Balooch, Noreen Zahra, Xiaolong Ren, Muhammad Rafiq, Muhammad Saqib

**Affiliations:** 1Institute of Environment and Ecology, School of Environment and Safety Engineering, Jiangsu University, Zhenjiang 212013, China; huangjiehp@ujs.edu.cn (P.H.); 2221809049@stmail.ujs.edu.cn (L.H.); 2College of Agronomy, Northwest A&F University, Yangling 712100, China or ch.sadam423@gmail.com (S.H.); or bilalhafeez32@gmail.com (M.B.H.); or rxlcxl@aliyun.com (X.R.); 3Department of Agronomy, University of Agriculture, Faisalabad 38040, Pakistan; sadamhussainuaf@gmail.com; 4Department of Botany, Ghazi University D.G, Khan 32200, Pakistan; Sidrabalooch820@yahoo.com or; 5Department of Botany, University of Agriculture, Faisalabad 38040, Pakistan; noreenzehrap@uaf.edu.pk or; 6Agronomic Research Institute, Ayub Agricultural Research Institute, Faisalabad 38040, Pakistan or rafiq_mdr164@yahoo.com (M.R.); or scsaqib@gmail.com (M.S.)

**Keywords:** antioxidants, chlorophyll, growth promotion, ionic balance, salinity stress

## Abstract

Seed priming with sorghum water extract (SWE) enhances crop tolerance to salinity stress; however, the application of SWE under salinity for camelina crop has not been documented so far. This study evaluated the potential role of seed priming with SWE in improving salt stress tolerance in camelina. Primed (with 5% SWE and distilled water-hydropriming) and nonprimed seeds were sown under control (no salt) and salt stress (10 dS m^−1^) conditions. Salinity reduced camelina’s emergence and growth, while seed priming with SWE improved growth under control and stress conditions. Under salt stress, seed priming with SWE enhanced emergence percentage (96.98%), increased root length (82%), shoot length (32%), root dry weight (75%), shoot dry weight (33%), α-amylase activity (66.43%), chlorophyll content (60–92%), antioxidant enzymes activity (38–171%) and shoot K^+^ ion (60%) compared with nontreated plants. Similarly, under stress conditions, hydrogen peroxide, malondialdehyde (MDA) content, and shoot Na^+^ ion were reduced by 60, 31, and 40% by seed priming with SWE, respectively, over the nonprimed seeds. Therefore, seed priming with SWE may be used to enhance the tolerance against salt stress in camelina.

## 1. Introduction

Camelina (*Camelina sativa* (L.) Crantz.), an important crop of family Cruciferae, is grown as a multipurpose crop. Camelina seeds and oil are used to produce biofuels [1], as animal feed [2], and as human feed (because it provides high levels of omega-3 fatty acid and alpha-linolenic acid) [3], as well as producing chemical derivatives that are used for paintings, coatings and cosmetics [4,5]. Camelina is an annual crop and has been cultivated since 4000 BC [6]. It is widely cultivated in Southeast Europe, Southwest Asia, the North and Central Plains, the United States and Canada [6,7,8,9,10]. Camelina is a native European species. In Europe, its cultivated area may exceed 10,000 hectares per year. It is cultivated in the vast areas of the world and often faces abiotic stresses (e.g., salinity, drought, and thermal stress, etc.) [11,12,13]. In many countries, soil salinization is becoming an increasing problem related to agriculture, affecting approximately 20% of the world’s irrigated arable land. In Pakistan, about 6.68 million hectares of land is affected by salinity, and about 40,000 hectares of land being degraded each year due to salt stress [14,15]. In China, it is estimated that 4.88% of the country’s total available land has been affected by soil salinization [16]. Salinity causes a significant reduction in growth attributes and photosynthetic activity of camelina plants [13]. High salt concentrations also cause a reduction in camelina yield by reducing the emergence percentage and growth attributes [17]. Soil salinity reduces the availability of water, caused ionic toxicity and osmotic stress [18,19], which results in a reduction in plant growth and development. The excessive production of reactive oxygen species (ROS) is deleterious to plants as it damages cellular processes [20]. The improvement of salinity tolerance in plants is considered an effective strategy for enhancing the crop performance, under saline environments [21].

Seed priming, a controlled hydration method used to induce a particular physiological state in plants by the treatment of the seeds with natural and synthetic compounds before germination, is a low cost and easy to adapt strategy for improving the growth and development of plants under environmental stimuli such as high salinity [22], drought [23] and thermal stress [24]. Seed priming enhances the emergence percentage, biomass production, crop productivity and balances the ionic homeostasis under salinity stress [25]. Allelopathy is a natural phenomenon in which different organisms affect the functioning of other organisms in their vicinity, negatively or positively, by releasing secondary metabolites, including phenolics, brassinosteroids, jasmonates, carbohydrates, and amino acids [26,27]: it is strongly linked with plant’s tolerance to abiotic stresses [28]. These secondary metabolites are present in different organs and play an important role in allelopathic effects [29]. When applied at low concentrations, allelochemicals have positive effects on crop performance [30,31,32]. However, it is hard to determine a definite mechanism for growth promotion due to the diverse nature and complicated interactions of allelochemicals under natural conditions. The previous studies have elucidated the positive role of secondary metabolites, hormones and some other natural compounds in plant growth promotion [31,32]. Allelochemicals affect other plants in different ways depending upon their concentrations. Major physiological processes like seed germination, seedling growth, chlorophyll accumulation, photosynthesis, transpiration, and leaf expansion are regulated by low concentrations of allelochemicals [27,31,32]. Furthermore, allelochemicals have an influence on hormonal balance, cell division, ion uptake and enzymatic activities. Secondary metabolites also have great potential for releasing P from the phosphates of metals like Al, Fe and Cu through their chelation. Growth is also promoted through optimum CO2 fixation, increased photosynthesis and stomatal conductance under normal conditions at relatively low concentrations of allelochemicals.

Seed priming with allelopathic water extract significantly improves the seed emergence, plant growth, and yield of various field gown crops, especially under environmental cues [33,34,35]. Under heat and drought stress, seed priming with allelopathic water extract improves the stay green, proline, and phenolic content, and stabilizes the biological membrane, thereby enhancing the plants’ tolerance to abiotic stresses [33]. Under salinity stress, seed priming with 5% aqueous sunflower extract can enhance the tolerance in rice cultivars [36]. Various allelochemicals have been identified in several plants, which can improve crop performance under normal and stress conditions. The exogenous application of allelopathic extracts may contribute to the production of endogenous secondary metabolites, which helps crops to perform better even under environmental cues [33]. The improved crop performance under exogenous application of allelopathic aqueous extract may be due to the increased accumulation of compatible solutes. In addition, under exogenous application of allelochemicals, the increased tolerance of crops may be due to improved water relations, enhanced photosynthesis, better absorption of water and nutrients, stomatal conductance, and antioxidant defense system [33,37,38,39,40,41]. When released from plants, allelochemicals may have to undergo some metabolic or environmental alterations in structure before they can exert biological activity. Allelopathic compounds exist in the form of seemingly disconnected structures and have different modes of action. The standard modes of release for allelochemicals are volatilization, residue decay, leaching, and root exudation. Sorghum is an important allelopathic crop that contains a diverse range of soluble phenolics [42], dhurrin, and cyanogenic glycoside, and has been reported to promote seed emergence and plant growth when its extract applied at low concentration [43]. The phenolic compounds (such as p-hydroxybenzaldehyde, vanillic acid, ferulic acid, p-hydroxybenzoic acid, and p-coumaric acid) also alter the mineral uptake, chlorophyll content, photosynthesis, carbon flow, and phytohormone activity. Phenolic compounds also inhibit the oxidation of the auxins induced by peroxidases and oxidases, and thus modulate the tissue auxin homeostasis. However, the active substances in sorghum water extract are highly dependent on the age and genotype of sorghum, location or environment, and cropping system. Under salt stress, seeds primed with sorghum water extract (SWE) exhibited more phenolic content, total soluble sugars and proteins, α-amylase activity, chlorophyll content, and K^+^ ion concentrations, thereby enhancing salt stress tolerance [44]. Under drought stress, foliar application of SWE significantly increased biomass production, chlorophyll and proline contents, and seed yield [45]. Sorghum extract is rich in phenolic acids and has antioxidant properties, and regulates plant growth and productivity under water-scarce conditions [46]. When applied at low concentrations, SWE also enhanced the tolerance of wheat crop to heat stress [47]. In recent years, the potential use of sorghum water extract (SWE) to enhance tolerance against abiotic stresses has been reported for different crops, such as maize (*Zea mays* L.) [48], wheat (*Triticum aestivum* L.) [44,49] and sunflower (*Helianthus annuus* L.) [30]. However, there is no study regarding the application of SWE for enhancing the performance of camelina under salinity stress. In this study, we hypothesized that seed priming with SWE would enhance the emergence and growth of camelina under salinity stress. This study aimed to estimate the effect of seed priming with SWE on the level of antioxidants, ROS activities, and uptake of essential nutrients in camelina under their association with salinity tolerance.

## 2. Results

### 2.1. Emergence Traits

Significant variations in the final emergence percentage and α-amylase activity of camelina were observed under the influence of salt stress and different priming treatments (Figure 1, Table 1, Appendix A). Seed priming with SWE significantly enhanced the final emergence percentage and α-amylase activity in salt treated camelina compared with nonpriming control and HP treatments. Under salt stress, the maximum emergence percentage was observed for seeds primed with SWE (64.33%), followed by hydropriming (46.67%). In a no-priming treatment, only 32.67% of seeds could germinate under salt stress. Under salinity, maximum α-amylase activity was observed for seed priming with SWE, which were 35.47 and 66.43% more than that of HP and NP, respectively. However, NP and HP treatments did not differ statistically under stress (Figure 1B).

FEP, final emergence percentage; RL, root length; SL, shoot length; RDW, root dry weight; SDW, shoot dry weight; Chl a, chlorophyll a; Chl b, chlorophyll b; H_2_O_2_, hydrogen peroxide; MDA, malondialdehyde; CAT, catalase; SOD, superoxide dismutase; POD, peroxidase; Na^+^, sodium ion; K^+^, potassium ion.

### 2.2. Seedling Growth

The data regarding the growth traits of camelina under salt stress and priming treatments are shown in (Figure 2). Salt stress hampered the seedling growth of camelina. Compared with the control (no salt), the root length, shoot length, root dry weight and shoot dry weight of salt-treated plants were decreased by 28.32, 23.27, 56.25, and 20.43%, respectively, across different priming amendments. Nonetheless, seed priming with SWE was effective in alleviating the salinity-induced inhibition of camelina growth. When compared with NP, under salinity stress, seed priming with SWE significantly increased the root length of camelina by 82.11%, shoot length by 33.34%, root dry weight by 75.02%, and shoot dry weight by 33.35% (Figure 2). Although HP was less effective than seed priming with SWE; it recorded significantly higher camelina growth than NP.

### 2.3. Chlorophyll Content

Salt stress severely decreased the chlorophyll content of camelina compared with the control. Nonetheless, seed priming treatments significantly (*p* ≤ 0.05) enhanced the chlorophyll content under salt stress, with the maximum increase in seed priming with SWE (Figure 3, Table 1). Compared with NP and HP, seed priming with SWE increased the chlorophyll a content by 72.70 and 26.66%, and chlorophyll b content by 85.18 and 35.13%, respectively. However, at salinity, SWE and HP treatments do not differ statistically for chlorophyll a content (Figure 3A).

### 2.4. Leaf H_2_O_2_ and MDA Contents

The results regarding the impact of salt stress and priming amendments on H_2_O_2_ and MDA content in camelina plants are depicted in (Figure 4, Table 1). In this study, lipid peroxidation was assessed in terms of MDA content in camelina leaves. The exposure of nonprimed camelina seeds (NP) to salinity stress significantly increased the H_2_O_2_ content, by 9.56 and 59.73%, and MDA content, by 61.32 and 82.562%, respectively, compared with HP and seed priming with SWE (Figure 4). However, HP and seed priming with SWE significantly assuaged the negative effects of salt stress. Compared with NP, the H_2_O_2_ content under HP and priming with SWE were reduced by 30.09 and 126.28%, respectively. The respective reduction for MDA content was 10.85 and 54.33%.

### 2.5. Antioxidant Enzymatic Activities

Salinity stress had a negative effect on the antioxidant enzyme activities in camelina seedling, except for catalase (CAT) in nonprimed seeds. However, priming amendments were effective in significantly increasing the antioxidant enzyme activities under salinity stress. A maximum increase in CAT activity was observed at salinity stress for seed priming with SWE compared with NP and HP. Likewise, superoxide dismutase (SOD) activity in salt-treated plants under seed priming with SWE was 33.80 and 10.46%, respectively higher than that of NP and HP. Seed priming with SWE and HP were the most effective treatments regarding the peroxidase (POD) activity and were statistically same (*p* ≤ 0.05) with each other under salt stress (Figure 5).

### 2.6. Mineral Ions Accumulation

Salinity stress markedly increased the Na^+^ ion concentrations in camelina shoots (Figure 6, Table 1). However, the priming amendments significantly (*p* ≤ 0.05) reduced the Na^+^ ion concentrations under salinity stress. Moreover, NP and HP treatments do not differ statistically for Na^+^ ions. Under salinity stress, the highest Na^+^ ions in camelina shoot (10.00) were observed for NP treatments. A variation among priming treatments for Na^+^ ion were also observed, and priming with SWE recorded significantly lowered (*p* ≤ 0.05) Na^+^ ion concentrations in camelina shoots. The K^+^ ion concentrations in the camelina shoots also differed statistically (*p* ≤ 0.05) in response to salt and priming treatments (Figure 6). Overall, exposure to salt stress reduced the K^+^ ion concentrations in camelina shoots. Nonetheless, the seed priming treatments significantly increased the K^+^ ion concentrations under control and salt stress conditions. However, at salinity, the NP and HP treatments do not differ statistically (Figure 6B). Compared with NP and HP, priming with SWE recorded 60 and 33.32% more K^+^ ion concentration under salt stress and was the most effective treatment for enhancing the K^+^ ions compared with NP and HP. 

## 3. Discussion

The main aim of this study was to recognize the salt-tolerant potential of camelina (*Camelina sativa*) at the emergence and seedling stages, and salinity adaptation was investigated by adopting the seed priming method with sorghum water extract (SWE) and hydro-priming (HP) separately. Under a high salt soil environment, the plants incurred a significant reduction in emergence percentage [50]; nonetheless, seed priming often mitigates salinity stress during emergence and subsequent crop growth. Although several seed priming studies provided a new insight into physiological changes under salt stress (37), the exact role of SWE in increasing emergence percentage and triggering early growth stages is still unknown. Allelopathic plants can be widely used in biofarming, considering their potential role in improving seed emergence. Salinity had a major inhibitory effect on the emergence, as shown in (Figure 1), which provides results for the two most important traits, such as emergence percentage and α-amylase activity. In this study, the emergence percentage and α-amylase activity were negatively affected under salinity stress, but very positively regulated by SWE followed by HP. It is noteworthy that both priming strategies (SWE and HP) increased the emergence percentage and α-amylase activity under control conditions and effectively reduced the negative consequences of salinity and improved the emergence attributes (Figure 1). In another case, Ahmad et al. [51] observed the reduction in time taken to 50% emergence and mean emergence time with SWE priming by using sorghum as a model plant. In this consistency, Li et al. [52] also found higher emergence percentages and α-amylase activity under various seed priming amendments. According to Ibrahim [53], seed priming involves prior exposure to abiotic stress, making a seed more resistant to future exposure. It also stimulates the metabolic processes and makes the seed ready for radicle protrusion. According to Bajwa et al. [44], priming *Triticum aestivum* (L.) with benzyl aminopurine (BAP) and SWE increased α-amylase activity and tissue K^+^ uptake under salt stress. Earlier, Jafar et al. [21] also reported a higher α-amylase activity in osmo-primed wheat plants under salt stress. Moreover, seed priming is also used to attenuate salinity stress for better growth and plant stand establishment. However, research regarding the effect of seed priming on seedling elongation is still contradictory [54,55]. In the present study, higher plant growth was observed while measuring the growth traits such as shoot length, root length, shoot dry weight, and root dry weight (Figure 2). Ahmad et al. [51] also reported the higher shoot length, shoot and root fresh and dry weight with seed priming with 1% SWE. This increase was quite related to the activation and synthesis of several enzymes, proteins, nucleic acids, and ATP build-up and effective repair of cytoplasmic membrane damage [55,56].

Interestingly, in camelina, the accumulation of shoot and root dry matter, obtained from unprimed seeds, was severely reduced; however, this effect was fully reversed under HP and seed priming with SWE treatments. In contrast, Karchegani et al. [57] observed that SWE suppressed the germination potential and seedling growth of milk thistle (*Silybum marianum* L.). Similarly, Khaliq et al. [58] also obserevd that SWE reduced the emergence rate, emergence percentage, and root and shoot length of *Cichorium intybus*, and suggested that this reduction might be due to the presence of allelochemicals in the SWE. The stunted plant growth mainly leads to a decrease in dry weight, as observed by several researchers [59,60]. The reduction in biomass may also be due to the accumulation of toxic ions [61] and increased respiration in response to salt stress [62].

The chlorophyll content is an effective index for plants to convert solar radiation into chemical energy. However, salinity stress negatively influences the photosynthetic efficiency (in terms of chlorophyll content) of camelina, as depicted in (Figure 3A,B). The reduction in photosynthetic rates in plants under salt stress is triggered primarily by reducing water potential and by the elevated levels of toxic ions (Na^+^ and Cl^−^) accumulated in chlorophyll and chloroplast that are the key regulators for plant health [63]. Simultaneously, the seed pretreatments help to improve the photosynthetic efficiency, as represented in (Figure 3). Na^+^ ion toxicity is not the primary reason for chlorophyll degradation under high salt concentrations, and an increase in Cl^−^ concentration may also lead to a decrease in chlorophyll content. Chloroplasts had higher permeability to Cl^−^, and NaCl treatments lead to the accumulation of Cl^−^ and the decrease in SPAD values. Therefore, under salt stress, leaf chlorophyll reduction may be related to a decrease in photosynthetic capacity. At high Na^+^ concentration, photosynthesis is mostly limited by stomatal conductance, which indicates that stomatal factors cause the reduction in CO_2_ assimilation. However, plants under Cl^−^ and NaCl treatments showed a reduction of C_i_: C_a_ (partial pressures of CO_2_ in the inside of the leaf and in the air) in parallel with a reduction in stomatal conductance, indicating specific damaging effects of Cl^−^ on the functioning of the chloroplasts in addition to stomatal limitations. High concentrations of Cl^−^ can be detrimental to the integrity of the cell and affect photosynthetic processes directly through membrane damage or enzyme inhibition if the vacuole can no longer sequester incoming ions. High Cl^−^ concentrations in the chloroplast also correlated negatively with the efficiency of Rubisco. In consistent to our results, Tahira et al. [64] also reported that seed priming significantly improved the chlorophyll content in *Hordeum vulgare* L.; this may be due to the smaller leaf area because under salinity stress, leaf expansion, associated with leaf anatomy changes (smaller and thicker leaves), is reduced, resulting in higher chloroplast density per unit leaf area. The protective effect of seed priming through enhancing the antioxidant activities and osmolyte production under stress conditions is also reported elsewhere [65]. In another study, Bajwa et al. [44] also found higher chlorophyll a, b, and total chlorophyll contents in *T. aestivum* with SWE priming (5% *v*/*v*) under salt stress. Kaur and Sharma [66] also reported that the application of allelochemicals reduced the chlorophyll biosynthesis, chlorophyll content, and fluorescence parameters and enhanced the chlorophyll degradation. In this study the photosynthetic apparatus was not affected with allelochemical priming, but rather photosynthesis was enhanced with SWE. Salinity-induced oxidative stress results in the excessive production of ROS (Figure 4A,B), which impairs the cellular functions by carrying out oxidative reactions with different biomolecules such as proteins, nucleic acid, and lipids, as well as inactivate enzymatic activities, and eventually leads to cell death [67]. Despite this, the seed priming treatments, HP and SWE, ameliorate salt-induced toxicity by scavenging ROS such as hydrogen peroxide to noninjurious level and control of MDA synthesis. Tahira et al. [64] observed a negative correlation of priming treatments with MDA production. The evidence revealed that one important aspect of seed priming, which lasts and remains on the plant after emergence, is the induction of nonenzymatic and enzymatic antioxidants, which helps plants to reduce the production of ROS, and to adapt under environmental cues [67]. In the current study, irrespective to salinity treatments, increased activities of CAT, POD and SOD were observed under priming treatments compared to unprimed plants (Figure 5A–C). Overall, ROS production was observed to be decreased in salt-treated plants compared to untreated plants; this may also be associated with the salt tolerance potential of camelina. However, Karchegani et al. [57] reported that the antioxidant activities, except for CAT, were significantly reduced in response to SWE or oven-chopped residual incorporation. A decraese in SOD and POD activities with allelochemicals has also been reported for wheat crops [68]. Additionally, ionic stress is an important factor of salinity-induced adversities, which are brought about by the excessive accumulation of Na^+^ ions, especially in the areal plant parts. Na^+^ ions compete and interfere with the K^+^ homeostasis, and its excess negatively affects the various physiological and metabolic processes in plants. Therefore, the Na^+^/K^+^ homeostatic is essential for better plant growth and development [69]. In this study, higher Na^+^ and lower K^+^ ion concentrations were observed for HP and seed priming with SWE (Figure 6A,B). Our study showed consistency with the findings of Rathod and Anand [70], who observed reduced Na^+^/K^+^ ratios in primed seedlings compared to unprimed ones. However, Rahneshan et al. [71] reported that salinity stress decraesed the nutrient uptake in a salt sensitive cultivar (pistachio), while it had no effect on the tolerant cultivar. The authors also noted that Na^+^/K^+^ levels were higher in the leaves of sensitive cultivar as compared to the tolerant one. Moreover, Bajwa et al. [44] also observed that priming *T. aestivum* (L.) with BAP and SWE reduced the Na^+^ ion concentration up to 5.78 and 28.3%, respectively, compared to nonprimed seedling, although the application of SWE as a priming agent is not fully understood and would be of great significance for plant stress physiology. Such studies may explore a new way of using allopathic plants to improve crop productivity by alleviating the adverse effects of environmental stresses. Therefore, an in-depth and comprehensive study is a must to draw definite conclusions on the results obtained in this study, with reference to the hormonal interplays and how they can be arbitrated by using different doses of SWE as priming agents.

## 4. Materials and Methods

### 4.1. Experimental Detail

The experiment was conducted in a greenhouse at the Department of Agronomy, University of Agriculture, Faisalabad during the July–August of 2020. During the growing season, an average temperature of 17.5 °C, relative humidity of 65.2%, sunshine of 7.2 h, and reference evapotranspiration (ETo) of 1.8 mm was observed. The seeds of the Camelina-618 line, having initial emergence >94% and 10.02% moisture content, were used for this experiment and collected from the Department of Agronomy, University of Agriculture, Faisalabad, Pakistan. Sandy loam soil (from Lyallpur soil series) with pH of 8.2, EC of 0.35 dS m^−1^, exchangeable sodium of 0.28 mmol/100 g, organic matter of 0.62%, nitrogen (N) of 0.05%, phosphorus (P) of 7.98 mg kg^−1^ and potassium (K) of 167.5 mg kg^−1^ were used for pot filling. The protocol described by Ryan et al. [72] was used for N, P, and K determination. 

The protocol of Cheema and Khaliq [73] was followed for the preparation of SWE. Dried sorghum plants were ground into small pieces and soaked in distilled water (1:10, *w*/*v* ratio) for 24 h. After filtration, the extract was labeled as 100% SWE, and a predetermined level of 5% SWE was prepared by diluting the 100% SWE [44]. Tap water with a pH of 6.8 was used for irrigation when required. The recommended dose of N (100 kg ha^−1^, from urea), P (30 kg ha^−1^, from diammonium phosphate), and K (60 kg ha^−1^, from muriate of potash) was applied at the time of pot filling.

### 4.2. Experimental Design and Treatments

In this experiment, seed priming treatments were: priming with sorghum water extract (SWE, 5%); and hydropriming (HP) and no-priming (NP) were kept as controls. The osmotic potential of applied SWE was recorded by an osmometer (Wescor Vapor Pressure Osmometer, VAPRO-5520) that was −0.781 MPa. The concentration of SWE was optimized based on camelina emergence and seedling growth performance during the preliminary laboratory experiment. For priming, camelina seeds were soaked in autoclaved distilled water (for hydropriming) and SWE (5%) with seed to solution ratio 1:5 (*w*/*v*), and kept in the dark for 24 h at 25 °C then dried to their initial weight. Salt stress (10 dS m^−1^, from NaCl) was imposed before two days of sowing following the method as described by Rhoades et al. [74]. Seeds were sown in plastic pots (29 cm × 18 cm), filled with 10 kg of well-sieved soil. The soil was collected from the Agronomic Research Farm Area, University of Agriculture, Faisalabad, and mixed well to reduce the soil heterogeneity. Twenty seeds were sown in each pot, and each treatment was replicated four times, and the experiment was laid out in a completely randomized design.

### 4.3. Observations

#### 4.3.1. Final Emergence Percentage

Seed emergence was counted daily, and the final emergence percentage (ratio of number of emerged seeds to the total number of seeds sown) was recorded by the method prescribed by the Association of Official Seed Analysts [75].

#### 4.3.2. Seedling Growth

After 20 days of sowing, plants were harvested for the estimation of seedling growth and biochemical analysis. Root and shoot length of ten randomly selected plants from each pot was measured by using a meter rod. The seedlings from each treatment were dissected into roots and shoots, and dry weights were recorded after oven-drying the samples at 75 ± 2 °C.

#### 4.3.3. α-Amylase Activity

In order to determine the α-amylase activity, a fresh seedling sample (0.1 g) was ground and mixed with distilled water (100 mL) and kept for 24 h at 4 °C. Then, the dinitro salicylic acid (DNS) based method described by Bernfeld [76] was used to estimate the enzyme activity by using the supernatant.

#### 4.3.4. Chlorophyll Contents Determination

The chlorophyll (Chl) contents were quantified according to the methods of Arnon [77]. The fresh leaves (0.5 g) were crushed by using a mortar and pestle, and extraction was done with 10 mL of acetone (80%). The extract was then left overnight in the dark. After filtration, the absorbance was taken through a spectrophotometer (model UV-4000) at the wavelengths of 645 and 663 nm. A blank with acetone (80%) was also run. The content of Chl a and Chl b were computed by using the given formulae.
(1)$$Chl\;a\;content\;\left(\mathrm{mg}/g\;fresh\;weight\right)\;=\frac{100\times \left(0.0127\times A663-0.00269\times A645\right)}{0.5}$$
(2)$$Chl\;b\;content\;\left(\mathrm{mg}/g\;fresh\;weight\right)=\frac{100\times \left(0.0229\times A645-0.00468\times A663\right)}{0.5}$$

#### 4.3.5. Antioxidants Defense System

SOD activities were measured by using the method of Giannopolitis and Ries [78]. The reaction mixture was prepared by mixing 20–50 µL enzyme extract, potassium phosphate buffer (pH 7.8), EDTA, riboflavin, methionine, and NBT. The test tubes were irradiated under a florescent lamp for 15 min, and absorbance was noted at 560 nm by using a spectrophotometer (UV-4000). The values were presented in mg g^−1^ FW by using the following formula:(3)$$\mathrm{Enzymes}\;\mathrm{units}\;(\mathrm{EU})=\frac{Enzyme*\left(light\right)-\left[Enzyme\#\left(light\right)-Enzyme*\left(dark\right)\right]}{Enzyme*\frac{\left(light\right)}{2}}$$
where: * without enzyme; # with enzyme

The prescribed method of Chance and Maehly [79] was followed for measuring the CAT and POD activities. Plant sample (0.5 g) was extracted in potassium phosphate buffer (50 mM). For CAT estimation, assay solution was prepared by mixing 0.1 mL enzyme extract, 5.9 mm H_2_O_2_, 50 mM phosphate buffer (pH 7.0), and a decrease in absorbance was detected at 240 nm after every 20 s.
(4)$$CAT\;activity=\frac{△A240\times Vt}{0.1\times V\times T\times W}$$
where: *V*, volume of the sample solution (mL); *Vt*, measured liquid volume (mL); *T*, measurement time; *W*, fresh weight of the sample

For POD activity, reaction mixture was prepared by mixing 0.1 mL enzyme extract, 40 mM H_2_O_2_, 20 mM guaiacol, and 50 mM potassium phosphate, and the absorbance was taken at 470 nm after every 20 s.
(5)$$POD\;Activity=\frac{△470\times Vt}{0.1\times V\times T\times W}$$
where: *V*, volume of the sample solution (mL), *Vt*, measured liquid volume (mL); *T*, measurement time; *W*, fresh weight of the sample.

#### 4.3.6. Determination of MDA and H_2_O_2_

The determination of MDA content was done by using the method of Carmak and Horst [80]. A leaf sample (0.5 g) was homogenized in 10 mL of trichloroacetic acid (TCA) (0.1%) solution, then the leaf extract was centrifuged for 15 min. The 4.5 mL of thiobarbituric acid (0.5%) was used for each mL of extract, and the reaction mixture was heated at 95 °C for 30 min. After cooling in ice bath, the samples were again centrifuged for 15 min. The absorbance was taken at 532 and 600 nm wavelengths and the concentration of MDA were calculated by using given formula:(6)$$MDA\;content\;\left(µ\mathrm{mol}\right)=\frac{\left(A\;532\;\mathrm{nm}-A\;600\;\mathrm{nm}\right)\times \Delta }{105\times 156}$$

A stands for the absorption coefficient (has value of 156 mm^−1^ cm^−1^).

According to Velikova’s method [81], the contents of H_2_O_2_ were determined. A sample of fresh leaves (2 g) was extracted with 10 mL of TCA (0.1%, *w*/*v*) at 450 °C and homogenized. After centrifuging of TCA containing extract, the 1 mL of supernatant containing 1 mL of sodium phosphate buffer (0.05 M, pH 7.0) and 2 mL of 1 M potassium iodide was used for the determination of H_2_O_2_. The absorption of the mixture was measured at the wavelength 390 nm by using spectrophotometer (model UV-4000), and water was used as blank. The values were expressed as mg g^−1^ FW.

#### 4.3.7. Na^+^ and K^+^ Ions Determination

Leaf Na^+^ and K^+^ concentrations were determined by using a flame photometer. Fresh leaves were digested using boiling deionized water, and the filtered solution was analyzed with a flame photometer for Na^+^ and K^+^ concentrations [82].

### 4.4. Statistical Analysis

The recorded data were shown as the means of four replicates. Analysis of variance technique was used by Statistix software (Statistix 8.1) for statistically analyzing the experimental data. The treatments were separated by using Least Significant Difference test at 5% probability level, and SigmaPlot version 10.0 (Systat Software, Inc., San Jose, CA, USA) software was used to draw the graphs.

## 5. Conclusions

Seed priming with SWE increased the emergence percentage and ionic homeostasis in Camelina under salinity, which lead to vigorous seedling growth, enhanced biomass production, and improved chlorophyll contents. It also resulted in the strengthening of the antioxidant defense system. Seed priming with SWE can thus be an effective technique for successful camelina production on salt-affected lands.

## Figures and Tables

**Figure 1 plants-10-00749-f001:**
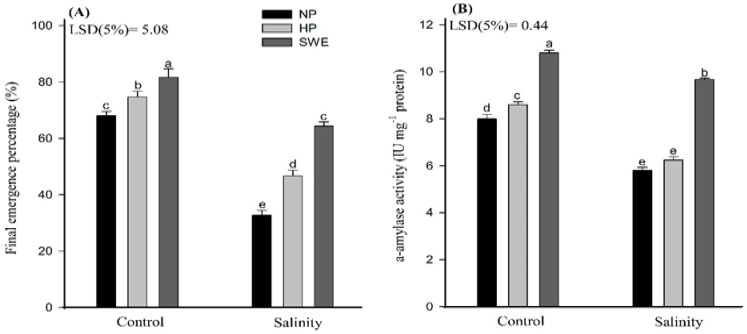
Influence of seed priming with sorghum water extract (SWE) on (**A**) final emergence percentage and (**B**) α-amylase activity of camelina under salt stress and control conditions. The α-amylase activity was recorded at 20 days after sowing (DAS). Error bar represents the standard error of four replications (*n* = 4). The means for stress treatments under each priming treatment not sharing a common letter are significantly different at the 5% level according to the LSD test. NP, no priming; HP, hydropriming.

**Figure 2 plants-10-00749-f002:**
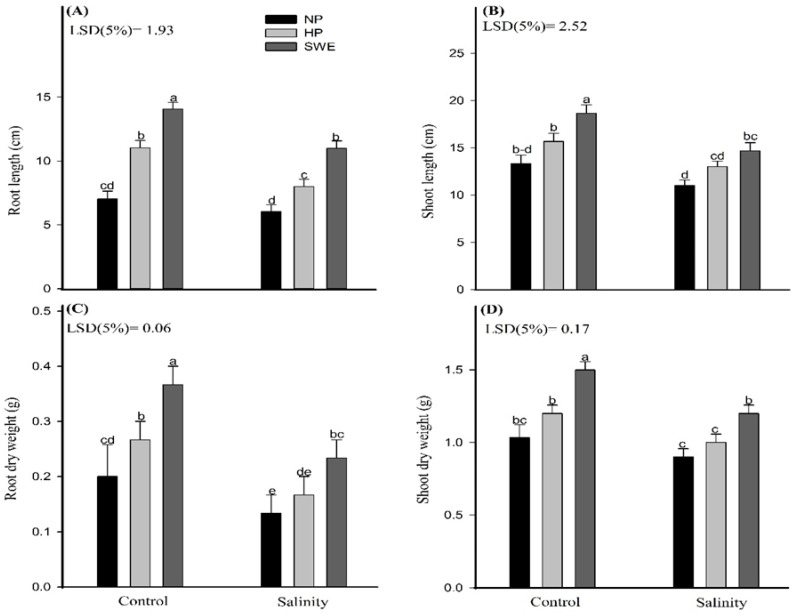
Influence of seed priming with sorghum water extract (SWE) on (**A**) root length, (**B**) shoot length, (**C**) root dry weight (g seedling^−1^), and (**D**) shoot dry weight (g seedling^−1^) of camelina seedlings under salt stress and control conditions. The data was recorded at 20 days after sowing. Error bar represents the standard error of four replications (*n* = 4). The means for stress treatments under each priming treatment not sharing a common letter are significantly different at the 5% level according to the LSD test. NP, no priming; HP, hydropriming.

**Figure 3 plants-10-00749-f003:**
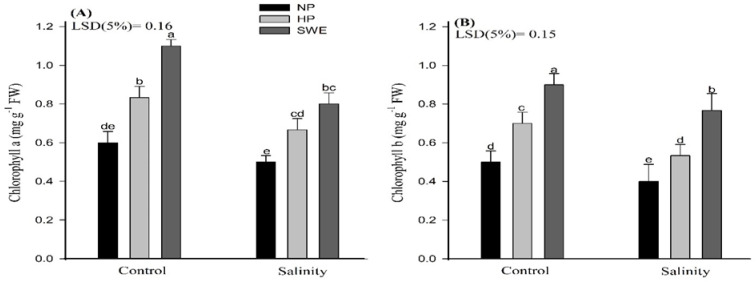
Influence of seed priming with sorghum water extract (SWE) on (**A**) chlorophyll a and (**B**) chlorophyll b content of camelina under salt stress and control conditions at 20 days after sowing. Error bar represents the standard error of four replications (*n* = 4). The means for stress treatments under each priming treatment not sharing a common letter are significantly different at the 5% level according to the LSD test. NP, no priming; HP, hydropriming.

**Figure 4 plants-10-00749-f004:**
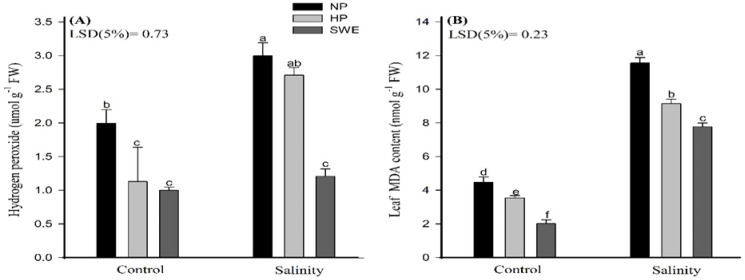
Influence of seed priming with sorghum water extract (SWE) on (**A**) hydrogen peroxide and (**B**) leaf malondialdehyde (MDA) content of camelina under salt stress and control conditions at 20 days after sowing. Error bar represents the standard error of four replications (*n* = 4). The means for stress treatments under each priming treatment not sharing a common letter are significantly different at the 5% level according to the LSD test. NP, no priming; HP, hydropriming.

**Figure 5 plants-10-00749-f005:**
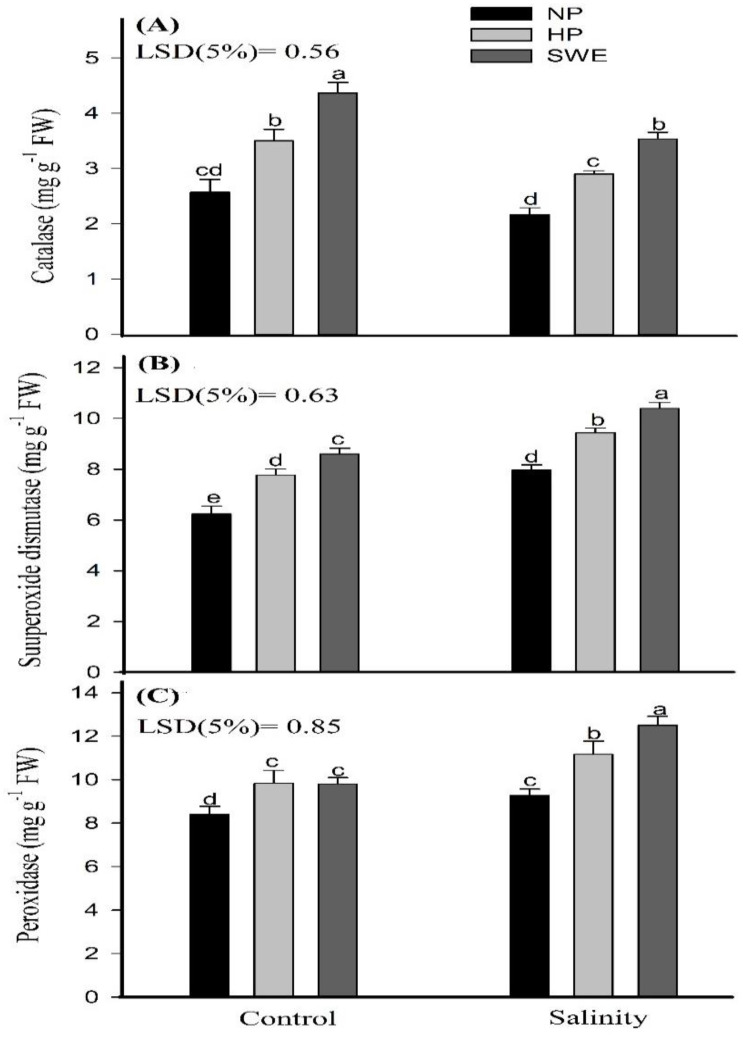
Influence of seed priming with sorghum water extract (SWE) on (**A**) catalyze, (**B**) superoxide dismutase, and (**C**) peroxidase activity in camelina seedling under salt stress and control conditions at 20 days after sowing. Error bars represent the standard error of four replications (*n* = 4). The means for stress treatments under each priming treatment not sharing a common letter are significantly different at the 5% level according to the LSD test. NP, no priming; HP, hydropriming.

**Figure 6 plants-10-00749-f006:**
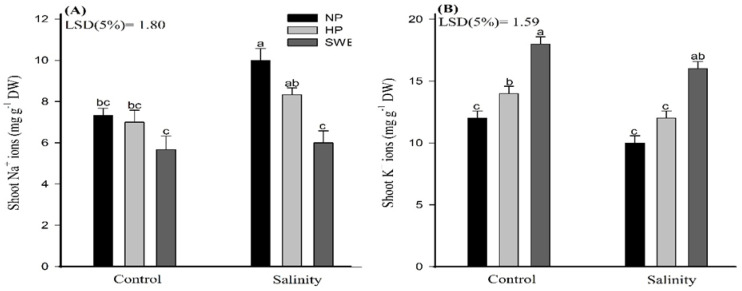
Influence of seed priming with sorghum water extract (SWE) on (**A**) sodium (Na^+^) and (**B**) potassium (K^+^) ions in camelina under salt and control conditions at 20 days after sowing. Error bar represents the standard error of four replications (*n* = 4). The means for stress treatments under each priming treatment not sharing a common letter are significantly different at the 5% level according to the LSD test. NP, no priming; HP, hydropriming.

**Table 1 plants-10-00749-t001:** Effects of sorghum water extract (control, hydropriming, and 5% SWE), salt stress (control and 10 dS m^−1^ from NaCl), and their interaction on the germination, morphological and physiological traits of camelina.

	**FEP**	**α-Amylase Activity**	**RL**	**SL**	**RDW**	**SDW**	**Chl a**	**Chl b**
Treatment (T)	99.09 ***	103.01 ***	17.39 ***	3.58 *	7.27 *	7.29 *	7.40 **	10.24 ***
Salt stress (S)	417.12 ***	156.06 ***	35.59 ***	10.43 **	11.36 **	17.41 **	24.07 ***	25.71 ***
T × S	15.75 **	287.53 ***	25.67 ***	18.15 ***	29.09 ***	20.66 ***	20.87 ***	14.52 ***
	**H_2_O_2_**	**MDA**	**CAT**	**SOD**	**POD**	**Na^+^**	**K^+^**	
Treatment (T)	13.53 ***	3281.34 ***	9.95 **	16.70 ***	7.58 *	3.60 *	18.26 **	
Salt stress (S)	4.42 *	3200.15 ***	34.11 ***	40.68 ***	25.54 ***	2.75 *	65.22 ***	
T × S	18.35 ***	1579.85 ***	21.41 ***	92.13 ***	51.95 ***	14.38 ***	33.91 ***	

The values are F-values showing *p* (* *p* < 0.5; ** *p* < 0.01; *** *p* < 0.001).

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
