# Peer review of "Seed Priming with Sorghum Water Extract Improves the Performance of Camelina (*Camelina sativa* (L.) Crantz.) under Salt Stress"

_plants, 2021, doi:10.3390/plants10040749_

Round 1

Reviewer 1 Report

The presented manuscript describes a currently much discussed topic to use plant extracts to alleviate the effects of various kinds of stresses on crops. In this manuscript the authors study salt stress. While the overall manuscript has been well written (as far as I can judge), I have some serious doubts on the experimental setup and statistical analyses used. If the carried out experiment is being repeated with exact same setup will the same results be reached? I am not convinced of that question.

methods
Was the used soil sterile? What about soil microbes? Was the same soil used for the treatments?
How many replications were used for each level? Were there any block effects?
Were the results weighted and compared to the control emergence of the seedlings? this is unclear!

introduction
The most often researched stress is heat stress. Please briefly explain the effects of sorghum extracts on other kinds of stresses than salt stress. What metabolites / natural products are expected to be found in sorghum water extract that may be responsible for the positive effects? Do some additional literature research.

l 33: human feed, please be more specific, what kind of food?
l 33: chemical derivatives, please be more specific, which substances?
l 39-44: directly mention physiological and molecular effects of salinity and give qualified literature references
l 47: please explain seed priming to reader's who don't know the term
l 49: allelopathy is not necessarily linked to chemical compounds, be more spcific and give examples
l 55: give examples, which compounds have which effects?

introduction fairly general, please be more specific and give examples directly related to your topic
are there any other studies that did the same? please compare

results
Fig 1: I prefer boxplots as the amount of variation is directly comparable between levels, please indicate the number of replications (n=??) in the plots for each level
Fig 2+3+4+5+6: see comments above

LSD test is not sufficient, I would prefer a linear model type of analysis, e.g. generalized linear model. An ANOVA with a post-hoc Tukey HSD would also be suitable. Prior to the analysis the authors need to verify normal (gaussian) distribution either with additional plots in the supplement or with Kolmogorov-Smirnov or Shapiro-Wilk test. Instead of comparing everything with everything, groups (NP, HP, SWE) should be compared only between control and salinity. This way, statistics are more evident.

discussion
are there any other studies that did the same? please compare and set in line with comparable studies
please also compare positive effects with other types of stresses

Reviewer 2 Report

Previously, some articles confirmed that seed priming with sorghum water extract (SWE) enhanced the crop tolerance to salinity stress. Authors transfer this idea on camelina. Primed with SWE (5%, hydroprimed and non-primed seeds were sown under salt stress and no stress conditions. Under salt stress, seed priming with SWE enhanced: emergence percentage, root length and dry weight, shoot length and its dry wieght, α-amylase activity, chlorophyll content, antioxidant enzymes activity and shoot K+ ion, and reduced the levels of hydrogen peroxide, malondialdehyde and shoot Na+ ion.  Authors concluded that seed priming with SWE may be used to enhance the tolerance against salt stress in camelina.

MAJOR POINTS:
1. Authors  followed too much (in my opinion) the article of Bajwa et al. 2018 titled:
“Seed priming with sorghum extracts and benzyl aminopurine improves the tolerance against salt stress in wheat (Triticum aestivum L.)”

1.1 By giving almost identical title:

Seed priming with sorghum water extract improves the tolerance against salt stress in Camelina [Camelina sativa (L.) Crantz.]

Bajwa AA, Farooq M, Nawaz A. Seed priming with sorghum extracts and benzyl aminopurine improves the tolerance against salt stress in wheat (Triticum aestivum L.). Physiol Mol Biol Plants. 2018;24(2):239-249. doi:10.1007/s12298-018-0512-9

1.2 By writing  the Introduction section, which seems in majority as a summary of introduction from the article Bajwa et al. 2018. Therefore, the introduction should be rewritten.

1.2. By performing identical analyses described before in Bajwa et al. 2018, however the latter did more analyses (including the effect of benzyl aminopurine).

  1. Authors stated in methods section that they used “The seeds of Camelina-618 line, having initial emergence >94% were used”.

However, results in Figure 1A show that non-primed (NP) seeds exhibited the emergence around 60% (no stress treatment). Additionally, SWE treatment did not increase the emergence to 90%. Please explain whether the seeds were aged? Stored at suboptimal conditions? They are categorized as orthodox, am I right? Please explain this declined emergence percentage.

  1. Authors stated that:
    “At salinity, maximum α-amylase activity was observed for seed priming with SWE, which was 35.47% and 66.43% more than HP and NP, respectively.”

However, Figure 1B shows that the activity in HP and NP  is similar and they do not differ statistically. Correct this misinterpretation.

  1. Figure 2 and results presented therein. The name of X axis cannot be “salt stress” because control conditions are not the salt stress. “Control” and “salinity stress” are sufficient.

Line 88, specify across which treatments.

Line 89, authors stated that “seed priming with SWE was effective in alleviating the salinity induced inhibition of growth.”

It is not a general true that salinity inhibited the growth of shoot or root. Not primed seeds under control and stress conditions were transformed in shoots and roots with identical length. See they are not significantly  different. Salinity stress did not cause a decrease in the length of root and shoot of camellia seedlings derived from not primed seeds. It must be precised.

  1. Figure 3 and results presented therein. The name of X axis cannot be “salt stress”. Explanation was given in the previous point.

Line 102, authors stated that “Salt stress severely decreased the chlorophyll content of camelina compared with the control.”

It is not true that salinity decreased the content of chlorophyll a. Its level is identical in non-primed seeds grown under control and salinity stress conditions. It must be precised.

Line 105, authors stated that “Compared with NP and HP, seed priming with SWE increased the chlorophyll a content by 72.70 and 26.66%,”

Add the information that the increase (26.66%) between SWE and HP treatment was not significant. It must be precised. Authors focused on data related to chlorophyll b and generalized about chlorophyll whilst data of chlorophyll a do not fit the description made by the authors.

  1. Figure 4 and results presented therein. The name of X axis cannot be here I andall following figures “salt stress”. Explanation was given in the previous point. Salinity stress affects roots. Authors did not show the results from roots. It is necessary to show them. Please consider that plant organs differentially react to salinity stress. Please read:

AbdElgawad H, Zinta G, Hegab MM, Pandey R, Asard H, Abuelsoud W. High Salinity Induces Different Oxidative Stress and Antioxidant Responses in Maize Seedlings Organs. Front Plant Sci. 2016 Mar 8;7:276. doi: 10.3389/fpls.2016.00276.

Line 114, authors stated that “The activities of ROS were assessed in term of H2O2 content”.

Precise that You measured one type of ROS instead  activity of the whole pool of ROS.

Line 116, precise whether seeds or seedlings were primed?

  1. Figure 5 and results presented therein.

Line 128, authors stated that “Salinity stress had a negative effect on antioxidant enzyme activities in camelina seedling”. It is not true in the case of catalase in non-primed seeds.

The units of enzyme activities are given in [mg g-1FW]. Please exlain how the results were calculated to obtain this unit in three enzymes.

  1. Figure 6 and results presented therein.

Line 143, authors stated that However, priming amendments significantly (P≤0.05) reduced the Na+ ion concentration under salinity stress”. According to statistics given in Figure 6, NP and HP are not different. Precise it.

Line 147, authors stated that “ K+ ion concentrations in camelina shoot were also significantly (P≤0.05) differed in response to salt stress and priming treatments (Figure 6).”

NP control and stressed seeds exhibited identical level of  K+. Precise it.

Line 148, authors stated that “  Exposure to salt stress reduced the K+ ion concentrations in camelina shoot.” It is true only for HP seeds. Precise it.

Line 149-151, except the HP under salt stress.

MINOR POINTS

Please define the status of this species in the world, is it still an oil plant grown in light soils or currently present as a weed in crops?

Discussion is short and should be improved to be more interesting for readers.

L.28 provide keywords different form these used in the title

L.80 define “DAS

L.166, it is partly known, for instance, the effect of sorghum phenolics to promote germination and growth is described in literature

L.194 “photosynthetic efficiency” was not measured

L.197 Explain how “ toxic ions (Na+ and Cl-) accumulated in chlorophyll”?

L.201 “this may be due to” smaller leaf area. Because under salinity stress, leaf expansion, associated with changes in leaf anatomy (smaller and thicker leaves), is reduced, resulting in higher chloroplast density per unit leaf area.

L.346 Italicize latin name

Reviewer 3 Report

With respect to the manuscript that I reviewed, you will be interested in the following appreciations and observations:

Abstract: concise, descriptive, emphasizing the idea, aim of the present study.

Introduction: very well written, well documented with relevant literature (very actual, from the last 5-10 years) and it is able to introduce and familiarize with the subject of the study, in a very gradual and logical way.

Results: logically presented including figure with 6 graphs regarding the camelina seedling variants response to saline stress.

Observation:

I would suggest changing the title of the figures (it is too long):

From: Figure 1. Final emergence percentage and α-amylase activity in primed and non-primed camelina seedlings under control (no salt) and salt stress condition. The α-amylase activity was recorded at 20 DAS. Error bar represents the standard error of four replications. The means for stress treatments under each priming treatment not sharing a common letter are significantly different at the 5% level according to LSD test.

In: Figure 1. Effect of salt stress on final emergence percentage and α-amylase activity in camelina seedlings at 20 DAS. Error bar represents the standard error of four replications. The means for stress treatments under each priming treatment not sharing a common letter are significantly different at the 5% level according to LSD test.

  • I think that the titles of the other figures can be modified in the same way. In addition, I believe that it would be good to explain at least the first graph what represents NPP and SWE.

Discussion: statements and conclusions are clearly supported by data and are linked to the paper's goal. Study implications and limitations are completely and succinctly presented.

To reformulate the phrase at rows 171-173, it is not clear.

Materials and Methods: procedures are clear, concise, and easily replicable.

Pay attention to some changes should be made:

-replace chaffed with ground (row 246)

- replace A-amylase with α- amylase (row 277)

- replace assy with assay (row 298)

- I think it must replace 4500C with 450C (row 314)

- delete (1M solution) (row 317)

- specify the calculation formula for the determination of H2O2 (row 319)

References: correlated well with the text

Reviewer 4 Report

Referee Report for the Manuscript ‘Seed priming with sorghum water extract improves the tolerance against salt stress in Camelina [Camelina sativa (L.) Crantz.]' submitted to the Plants by Adeel Abbas, Ping Huang, Sadam Hussain, Saddam Hussain, Daolin Du1, Muhammad Bilal Hafeez, Sidra Balooch, Noreen Zahra, Xiaolong Ren, Muhmmad Rafiq and Muhammad Saqib

The manuscript analyses the potential of sorghum water extract in Camelina salt stress tolerance regulation.
Potentially, the manuscript could provide an important contribution to the research area.
It has a certain potential to be published in the Plants.
However, the MS should be still optimized significantly to be more consistent with the science and the high level of the Plants.
I recommend accepting this manuscript after a major revision only.

Comments, questions, and recommendations to this manuscript:

1. The abstract should be revised substantially and conceptually improved.

2. What are the areas of saline soils in China and Pakistan? What are salinization types in both there?

3. A more detailed discussion of the revealed changes in amylase activity is needed in the context of the investigation.

4. A more detailed discussion of the detected changes in chlorophyll is needed in the context of the study.

5. A more detailed discussion of antioxidant enzyme changes in the context of this work is needed.

6. A more detailed discussion of the K changes in the context of the study is needed.

7. The authors consider 'seed priming' as 'a controlled hydration method'. More details about this are needed.

8. The authors consider priming in the context of a process to prepare plant 'primed state' for future stress or as seeds treatment/seed dressing?
It is necessary to substantiate it more thoroughly in the context of the research concept in the manuscript.

9. There is a need to achieve a greater level of conceptualization, contextualization, and consistency in the manuscript.

10. The conclusion should be made more detailed, reasoned.

11. How would the authors classify sorghum water extract? As plant growth regulator? Biostimulant? Fertilizer? Improver? Antistress agent? Something else?

In any case, definitely, it is necessary to determine the appropriate terminology and categorization, to provide and substantiate a definition supported by the authors of the manuscript in the context of the scientific and regulatory definitions existing.
In this context, there are fundamental 'classical' and highly conceptual works considering these types of bioproducts:
du Jardin, P. (2012). The Science of Plant Biostimulants - A Bibliographic Analysis, Ad hoc Study Report. Brussels: European Commission.
du Jardin, P. (2015). Plant biostimulants: definition, concept, main categories and regulation. Scientia Horticulturae,196, 3-14.
Yakhin, O. I., Lubyanov, A. A., Yakhin, I. A., & Brown, P. H. (2017). Biostimulants in plant science: a global perspective. Frontiers in plant science, 7, 2049.
Caradonia, F., Battaglia, V., Righi, L., Pascali, G., & La Torre, A. (2018). Plant Biostimulant Regulatory Framework: Prospects in Europe and Current Situation at International Level. Journal of Plant Growth Regulation, 1-11.

12. It is desirable to provide more discussion about the active substances of the sorghum water extract as well as the potential of their interactions in the context of hypothesized mode and mechanism of action to the plant.

13. By the way, how do authors distinguish between concepts 'mode of action' and 'mechanism of action' or these are similar in the context above?

14. In the contexts above, it is also necessary to discuss the concepts of the potential of additivity, synergism, and emergent properties of the components in the manifestation of the sorghum water extract’s mode/mechanism of action.

15. What about environmental and toxicological characteristics, physical and chemical properties of the sorghum water extract in more detail?

16. It is necessary to make more sure that this is indeed the first study with sorghum water extract in the context of the manuscript goal and tasks.

17. To exclude the potential risk of conceptual plagiarism and self-plagiarism, it is necessary to look once again through all close to this manuscript and other publications.
Provide necessary appropriate references and discuss earlier results, if so.
Authors' and other papers from any sources, and languages should be considered.

18. What principally innovative concepts have been suggested by authors?
What is principally new this MS provide?

Round 2

Reviewer 1 Report

In the first revision the authors added some sentences and paragraphs in the introduction and discussion to highlight the major aspects of salinity and their results. In the discussion, the authors need to add more qualified references that backup their results, especially the section on Na+. Salinity is a well-studied phenomenon in plants and the authors need to put their results in line with other published studies. Contradictory results are also very interesting and should not be forgotten if they have been found.

In Table 1, the authors present p-values. Please add a Supplement and provide the output of the statistical model used, including validation metrics (e.g. AIC / BIC and F- / p-values for the individual factors and levels). I would also like to see the same output for all of the Figures. These metrics are important to evaluate the soundness of the results. 

Lastly, although I don't feel qualified enough to jugde on the English, I would recommend extensive editing of english by a native speaker.

Author Response

"Please see the attachment

Reviewer 2 Report

Authors intensively worked on te manuscript and improved it significantly.
I recommend to accept it. Several editorial mistakes related to non-italicized species name in references  1, 13, 41, 73 can be updated at the manuscript production stage.

Author Response

"Please see the attachment

Reviewer 4 Report

1. Maybe it would be more correct to talk about 'seed treatment' / 'seed dressing' and 'priming effect' instead of 'seed priming'?

2. What is your evidence that the Sorghum water extract is a plant growth regulator?
Clarify your real arguments.
According to the literature cited in the first referee report, it is definitely biostimulant.
There is a difference.

3.  The difference between 'mode of action' and 'mechanism of action' is not substantiated enough.

4. The concepts of the potential of additivity, synergism, and emergent properties of the components in the manifestation of the sorghum water extract’s mode/mechanism of action are not substantiated enough.

5. The manuscript should be revised and conceptualized more.

Round 3

Reviewer 1 Report

All my comments have been addressed

Author Response

We thank you and the reviewers for useful comments and suggestions, which have significantly improved the manuscript (plants-1097915). We have revised the manuscript and incorporated of all the suggestions. 

Reviewer 4 Report

Definitely, changes made by authors in response to questions and comments from the reviewer are superficial.
The challenges of 'seed priming',  'sorghum water extract as a plant growth regulator or biostimulant',  'mode of action' and 'mechanism of action', 'additivity, synergism, and emergent properties' are not disclosed and not sufficiently discussed.
This manuscript does not provide sufficient background and all necessary and relevant references.
In its current form, the manuscript does not at all correspond to the high level of the target journal.
This manuscript has the potential for greater conceptualization and contextualization as a whole, including the need for more in-depth analysis and system approach there, and should provide all these.
It is necessary to revise the manuscript again, taking into account earlier questions, remarks, and comments of the reviewer and significantly improve the manuscript.

Author Response

Please see the attachment."
